# Lateral Flow Immunoassay Reader Technologies for Quantitative Point-of-Care Testing

**DOI:** 10.3390/s22197398

**Published:** 2022-09-28

**Authors:** Jongwon Park

**Affiliations:** Department of Biomedical Engineering, Kyungil University, Gyeongsan 38428, Korea; jpark3@kiu.ac.kr; Tel.: +82-53-600-5721

**Keywords:** assay quantification, colorimetric reader, dual-signal reader, electrochemical reader, fluorescence reader, lateral flow immunoassay, magnetic reader, photothermal reader, point-of-care testing

## Abstract

Due to the recent pandemic caused by coronavirus disease 2019 (COVID-19), the lateral flow immunoassay used for its rapid antigen test is more popular than ever before. However, the history of the lateral flow immunoassay is about 60 years old, and its original purpose of use, such as a COVID-19 rapid antigen test or a pregnancy test, was the qualitative detection of a target analyte. Recently, the demand for quantitative analysis of lateral flow immunoassays is increasing in various fields. Lateral flow immunoassays for quantitative detection using various materials and sensor technologies are being introduced, and readers for analyzing them are being developed. Quantitative analysis readers are highly anticipated for their future development in line with technological advancements such as optical, magnetic field, photothermal, and electrochemical sensors and trends such as weight reduction, miniaturization, and cost reduction of systems. In addition, the sensing, processing, and communication functions of portable personal devices such as smartphones can be used as tools for the quantitative analysis of lateral flow immunoassays. As a result, lateral flow immunoassays can efficiently achieve the goal of rapid diagnosis by point-of-care testing. Readers used for the quantification of lateral flow immunoassays were classified according to the adopted sensor technology, and the research trends in each were reviewed in this paper. The development of a quantitative analysis system was often carried out in the assay aspect, so not only the readers but also the assay development cases were reviewed if necessary. In addition, systems for quantitative analysis of COVID-19, which have recently been gaining importance, were introduced as a separate section.

## 1. Introduction

The lateral flow immunoassay (LFIA) has advantages such as reasonable sensitivity and specificity, rapid time-to-result readout, ease of operation, no need for a readout device for qualitative analysis, and low cost. For these reasons, it has been used in various fields such as disease diagnosis, consumer protection, and environmental monitoring for about 60 years [1,2,3,4]. Traditionally, the LFIA has been used as a tool for qualitative measurements and is performed by determining whether a test line has formed through the user’s eye. The structure of a general LFIA is shown in Figure 1. The assay is usually placed inside a plastic housing, which consists of several different functional pads and a porous membrane. There is a backing support at the bottom of the assay. First, the liquid sample is introduced through the sample well of the housing and the sample pad of the assay. It then flows into the conjugate pad, where primary binding of the target analyte and the labeled immunoreagent occurs. Afterwards, the sample is developed through a porous nitrocellulose membrane, where secondary binding of the target analyte and immunoreagent occurs to form a test line, and another binding independent of the target analyte is induced to form a control line. The control line is used to ensure that a sufficient sample has been developed for the test, otherwise the test will be invalidated. Finally, the sample flows into the absorbent pad and the result can be checked through the reading window of the LFIA housing [3,4].

An LFIA for qualitative measurement has the following disadvantages: it is only used for primary screening, there is no automatic test result documentation, there is a high possibility of false positives and false negatives due to technical errors in sample handling, it has limited multiplexing function, and there are limitations of high-throughput diagnostics due to manual operation [3,5]. Many studies have shown that some or a significant part of the above-mentioned shortcomings can be overcome by using a quantitative immunoassay system consisting of an assay applied with biochemical technology and a reader with various sensing technology. In the future, the demand for quantitative analysis systems is expected to increase due to the following reasons: increasing demand to convert tests that were only possible in the central laboratory to on-site diagnosis, the need to feedback test results to other healthcare systems, and an increased need to obtain analytical results that are not visible to the naked eye.

Readers for quantitative analysis of LFIA can be classified into several types as follows according to the used label and its measurement technology: (1) colorimetric readers that measure the contrast or color change of colloidal gold or colored monodisperse latex particles using a charge-coupled device (CCD) or a complementary metal–oxide–semiconductor (CMOS) camera sensor; (2) fluorescence readers that excite fluorescent monodisperse latex particles using light sources such as light-emitting diodes (LEDs) and measure their luminescence intensity with an optical sensor such as a CCD or CMOS; (3) magnetic readers that measure the magnetic field intensity of paramagnetic monodisperse latex or superparamagnetic iron oxide particles using magnetic field sensors such as a giant magnetoresistance sensor; (4) photothermal readers that measure the heat of label particles by light excitation; (5) electrochemical readers that measure changes in electrical properties such as voltage, current, and impedance appearing on label particles; and (6) dual-signal readers that use two or more simultaneously in the measurement principle of (1)~(5) mentioned above [6,7]. Table 1 compares the label particles, output signals, and measurement sensors used in each reader.

In this review, research and commercialization cases of LFIA readers for quantitative analysis were reviewed for each type of system described above. In addition, the quantitation reader recently used in the diagnosis of coronavirus disease 2019 (COVID-19) was also reviewed. The trend analysis or performance comparison described for each type of reader will be helpful for researchers conducting related research in selecting an appropriate quantitative analysis method and verifying the performance of the developed system.

## 2. Colorimetric Readers

CCD or CMOS image sensors have been used for the quantitative analysis of LFIAs using color-intensity-varying label particles such as colloidal gold or colored monodisperse latex. As shown in Figure 2, such a colorimetric reader consists of mechanical parts that can store LFIA, a broadband light source that illuminates the test line and control line of the assay, a sensor that acquires images of the region of interest (ROI) including the test line, and a processor with image-processing software that calculates the presence or concentration of a target analyte from an image. The quantification performance can be improved by using an optical bandpass filter that only accepts images in a specific color band. Sometimes, this is implemented as part of image processing by software without the use of a physical filter.

The image sensor mounted on the reader acquires an image of the test line, which is a combination of labeled particles, antigens, and antibodies. The pixel intensity of the test line is then converted to the concentration of the target analyte. The colorimetric LFIA reader is most widely used for assay quantification because of its simple structure and small size, but problems such as high probability of false positives and false negatives and limitations in accurate and multiple quantification are pointed out [8,9].

Park developed an optimized colorimetric reading method to improve the sensitivity of the colorimetric LFIA reader for quantitative analysis. Quantification of LFIA was performed using a color camera mounted on a reader and Bayer filter images obtained from it. Among Bayer-filtered images such as red, green, and blue, the image that was most sensitive to the target analyte concentration was used to improve the performance [10]. In order to improve the sensitivity of the optical reader, there is also a case of using light illumination with a color of a specific wavelength band rather than the commonly used white light. Based on the fact that the complementary color of the gold particles used in their LFIA is green, Zeng et al. used a green LED with a center wavelength of 540 nm as the light source for the reader. An automated process by applying a fuzzy cellular neural network with adaptive threshold (FCNN-AT) algorithm was developed for noise reduction of the obtained assay image and image segmentation of ROI regions (i.e., test line and control line regions). The degree of improvement was evaluated by comparing peak signal to noise ratio (PSNR) and contrast of between-class variance (CBCV) values before and after application of the developed algorithm. These values are indicators of image quality and image segmentation performance, respectively [11]. There is also a case study confirming that the sensitivity of the LFIA quantitative analysis reader can be improved by 3 to 5 times by manually adjusting the exposure time of the camera that acquires the assay image [12].

The number of research cases that quantify LFIA using cameras, LED lighting devices, and image-processing applications mounted or installed on smartphones is also increasing. The use of mobile devices as an in vitro diagnostic device for quantitative analysis not only uses the high-performance hardware and software installed in it, but also because it was originally created for communication purposes, the measurement results can be quickly and simply communicated with healthcare facilities such as hospitals. As an example, an LFIA for environmental monitoring was developed to measure Hg(II) using a gold-nanoparticle-based streptavidin-biotinylated deoxyribonucleic acid (DNA) probe. Hg(II) was quantitatively measured with a smartphone camera and a written smartphone application, and the limit of detection (LOD) was 2.53 nM [13]. Lee et al. developed a portable platform called ‘NutriPhone’ for quantitative analysis of blood vitamin B12 consisting of LFIA, application software, and smartphone accessories as shown in Figure 3. In LFIA, a spacer pad was used to improve the test efficiency by delaying the sample deployment time. A Gaussian filter and median filter were used to improve the signal-to-noise ratio of the obtained assay image and to convert the 2-D image into 1-D array data, respectively [14]. Ruppert et al. developed a smartphone-based detection system including LFIA and measurement software for quantitative measurement of digoxigenin. The performance of digoxigenin quantification (i.e., LOD, limit of quantification (LOQ), and limit of blank (LOB)) using the developed system was compared with those using other reference methods to confirm its effectiveness [15]. There is also a research case in which an ambient light sensor of a smartphone, a component originally used for the purpose of adjusting the brightness of a smartphone screen, was used as a sensor for quantitative analysis. In this study, a mechanical and electrical adapter including a light source utilizing LED was designed and manufactured using a 3-D printer [16]. Yu et al. developed a disposable LFIA that could quantitatively measure alkaline phosphatase (ALP) activity in milk using a smartphone camera. The LOD was 0.1 U/L, and the linear dynamic range was 0.1–150 U/L [17]. In addition, Zhang et al. used a smartphone as a reader device for quantitative analysis of colloidal gold label particle-based assays and developed related software. The performance of quantitative analysis was improved by applying the grayscale projection value processing algorithm in recognizing the ROI in the obtained assay image. As a result of quantitative analysis of human chorionic gonadotropin (hCG) using the developed system, the LOD and linear ranges were 3 ng/mL and 6–300 ng/mL, respectively [18].

In a quantitative analysis system that performs colorimetric reading of latex particles, there is a commercialization case in a unique form that integrates an assay and a reader into one device. In 1987, Swiss Precision Diagnostics GmbH developed a pregnancy test kit using blue latex particles as a label, which is still being sold. The most unique feature of this product is that optoelectronics (i.e., sensors, LEDs, battery, etc.) that measure the intensity of the test line are integrated into the assay device as shown in Figure 4. The intensity of the test line is compared to a preset threshold, and the pregnancy status and post-fertilization time are printed on the black and white liquid-crystal display (LCD) window integrated in the kit. The latest product displays not only pregnancy status, but also how many weeks have passed since conception in words [19].

The first type of reader for quantification of LFIA was the colorimetric reader and is still the most widely used reader. Therefore, in some cases, general-purpose commercial readers are used for quantitative analysis of LFIAs developed by researchers. Liu et al. quantitatively analyzed chromium ions in water and serum samples using the AESE-Quant Portable and Benchtop Lateral Flow Reader (DCN Inc., Irvine, CA, USA). The linear detection range of the measurement was 5–80 ng/mL [20].

The points to be considered in designing a colorimetric reader are as follows: (1) illumination and lens effects must be considered, (2) positioning errors should be recognized and compensated, (3) the appropriate choice should be made between off-axis measurement or confocal measurement, and (4) it should be reviewed whether the specifications of the optical sensor are appropriate.

## 3. Fluorescence Readers

Figure 5 is a conceptual diagram of the main components of a fluorescence reader. As shown here, most of the components constituting colorimetric readers can also be used for the quantification of LFIA using fluorescent labels that emit light of wavelengths in visible or infrared bands. However, in this case, since the image sensor measures the emission of label particles rather than reflected light, it can be expected to improve quantitative analysis performance [7]. For excitation of fluorescence labeling particles, a light source with a specific narrow wavelength band is used, and light emission is also within a narrow wavelength band, so optical bandpass filters are used more often than in colorimetric readers.

Berlina et al. developed a fluorescent LFIA for quantitative analysis of chloramphenicol in milk. A quantum dot with a peak emission of 625 nm wavelength was used as a label particle, and an LOD of 0.2 ng/mL was obtained in quantitative analysis using a photometer [21]. It is also possible to simultaneously detect multiple analytes by using the characteristics of a fluorescent label particle with a narrow emission wavelength band. Lu et al. developed a platform for rapid and simultaneous quantification of iron (ferritin), vitamin A (retinol-binding protein), and inflammation status (C-reactive protein) using multiple fluorescent label particles as shown in Figure 6. A reader using a CMOS sensor called ‘TIBIT’ was produced and smartphone application software for image processing called ‘NutriPhone’ was developed [22]. The developed reader was later utilized to measure progesterone in cattle [23], and the application software was applied to a colorimetric portable platform for quantitative analysis of vitamin B12 in blood [14].

The above study is to quantitatively analyze two or more measurement target substances by placing multiple test lines in one assay. There are also research cases where one assay has a test line for measuring one analyte, and several assays for detecting different analytes are configured in the form of a cartridge. Rong et al. designed and fabricated test cartridges for automated sample fluid delivery and multiplex detection. The centrifugal force obtained by rotating the cartridge was used to simultaneously develop the sample fluid on multiple strips at a speed of about 1 mm/s. In addition to the assay cartridge, a miniaturized standalone strip reader was manufactured and using this, human immunodeficiency virus antibody (HIV Ab), treponema pallidum antibody (TP Ab), hepatitis C virus antibody (HCV Ab), and hepatitis B virus surface antigen (HBsAg) were simultaneously quantified. The LODs were 0.11 NCU/mL, 0.62 IU/L, 0.14 NCU/mL, and 0.22 IU/mL, respectively [24].

When Raman scattering occurs, the electric field of scattered light resonates with a specific metal material, and this resonance enhances the Raman scattering signal by about 100 times to create an electric field with large amplitude. An analysis method using this phenomenon is called surface-enhanced Raman spectroscopy (SERS). Currently, LFIA using this principle and gold nanoparticles as label particles is being actively developed. However, general SERS-based readers are expensive and have a disadvantage in that multi-detection is difficult [25]. To solve these problems, a portable SERS-based lateral flow immunoassay reader was developed, and the developed system used an integrated LFIA reaction column for multiple detection. Using the designed reader system, cancer markers such as alpha-fetoprotein (AFP), carcinoembryonic antigen (CEA), and prostate-specific antigen (PSA) could be detected with a detection limit of 0.01 ng/mL [9]. Tran et al. developed a portable Raman/SERS-LFIA reader to shorten the time required for SERS reading (i.e., scan) and make it suitable for on-site diagnosis. The developed reader is equipped with a fiber optical device that can illuminate the entire width of the assay line as shown in Figure 7, and using this, a reading time of 5 s was realized. In the quantification of hCG using this, it was possible to measure up to 1.6 mIU/mL [26].

The method of collecting a target analyte using magnetic nanoparticles (MNPs) and attaching labeling particles to it has its own advantages, but it has problems such as poor particle stability, dispersion due to aggregation, and sinking during the treatment and measurement process. A laser-induced fluorescence (LIF) reader was developed that used the ratio of fluorescence measurement to absorbance measurement using a turbidimetric system to quantify the analyte. Although it is not a case applied to the LFIA format, the developed reader was applied to the measurement of salinomycin concentration and showed an LOD of 39 pg/mL, and the linearity and sensitivity of quantitative analysis were improved [27].

In order to improve the quantitative analysis performance of assays using fluorescent particles as labels, time-resolved fluorescence LFIA (TF-LFIA), which does not simply measure the intensity of light emission but observes light changes over a short period of time, is also being developed. The fluorescence label particles used in TF-LFIA emit short-lived or long-lived fluorescence when excited by a temporally short pulse of light using a light source such as a laser. TF-LFIA, which measures the change in light over time, is more efficient than steady-state fluorescence intensity detection because it can reduce background noise and increase measurement sensitivity [28]. Lu et al. developed TF-LFIA for the measurement of brucellosis using a fluorescent europium nanospheres-protein G (EuNPs-prG) label and purified lipopolysaccharide (LPS) capture antigen. A commercial reader (AFS-1000, Guangzhou Lanbo Biotechnology Co., Ltd., Guangzhou, China) was used as a quantitative reader. The linear dynamic range was 1.6–100 IU/mL, the lower LOQ was 1.6 IU/mL, and the LOD was 0.3 IU/mL [29].

Similar to the colorimetric reader, there are cases in which a smartphone is used as a reader in the quantification of an assay using a fluorescent label. In this case, since light with a specific wavelength band is required for excitation of the label particle, it is designed separately, but a smartphone is generally used for light reception and signal processing. Jin et al. simultaneously quantified mercury ions, ochratoxin A, and Salmonella using an LFIA using aptamer-functionalized multi-colored upconversion nanoparticles and a smartphone-based portable device. LODs were 5 ppb, 3 ng/mL and 85 CFU/mL, respectively [30,31].

Recent state-of-the-art in vitro diagnostic systems, including LFIA systems, use excellent sensor technology to improve the accuracy, precision, and linearity of the measurement. However, there are still the problems of quality control that arise from failing to reflect change factors such as performance change for each batch of assay production or reader sensor drift due to long-term use. There are also a research cases on the technical design to solve these quality control problems of the fluorescence reader. In addition to labeling particles on the test line, Huang et al. prepared another microsphere to capture for internal quality control. Quantitative analysis of glycosylated hemoglobin type A1c (HbA1c) was performed using the ratio of luminescence intensity from labeling particles and quality control particles. When using the conventional method without quality control particles, the coefficients of variations (CVs) of high and low HbA1c were 4.65% and 4.08%, respectively. On the other hand, when particles were additionally applied for internal quality control, they were improved to 0.69% and 1.05%, respectively [32].

One of the well-known systems in the commercialization of assays using fluorescent labels and readers is Alere’s Triage^®^ system (Waltham, MA, USA), shown in Figure 8. The Triage^®^ System is an immunoassay system for quantitative analysis of multiple analytes in blood or urine. The Triage^®^ Meter is used for quantitation of the assay. It includes several internal controls, external controls, and quality control (QC) lock functions. A fluorescent energy transfer dye system was used for labeling the target material. The excitation wavelength was 670 nm, and the emission wavelength was 760 nm. In the Triage^®^ Meter, a laser diode was used as a light source for excitation of the label, and a photodiode was used as a sensor for measuring the light from the dye. Calibration curves for quantitative analysis and software for meter operation are downloaded to the instrument through a code chip or electrically erasable programmable read-only memory (EEPROM). In addition, the reader is calibrated by measuring the gold-standard fluorescence energy transfer latex (FETL) inside the Triage^®^ Meter. This calibration is performed for every measurement using the assay and at power-up of the meter. The assay is calibrated for each lot using the measurement target material whose concentration is known by the manufacturer, and the calibration data are transmitted to the user through the code chip enclosed for each assay lot and downloaded to the user’s device. The commercial success of the Triage^®^ system can be attributed to the application of these meticulous analysis-specific and reader-specific calibration algorithms and improved user-friendliness [33].

Up-converting phosphor is a crystalline material synthesized from rare earth metals and is excited by infrared light to emit light in the visible wavelength band. Compared to conventional reporter particles such as colloidal gold or colored latex, it is evaluated to have excellent sensitivity and high stability [34,35,36]. Yan et al. developed LFIA using up-converting phosphor particles as labels and quantitatively analyzed Yersinia pestis. The reader for quantitative analysis excited the label using an infrared (IR) laser with a wavelength of 980 nm and converted the visible luminescence with a wavelength of 541 nm into a voltage using a photo multiplier tube (PMT). The reader used in this study obtained a voltage profile according to points on the strip with a fixed PMT while moving the strip, and the maximum value of the obtained voltage profile was used for the quantification of analytes. The LOD was 10^4^ CFU/mL [37].

## 4. Magnetic Readers

Magnetic particles can be used as a label for the LFIA, and in this case, quantification of the captured magnetic label is performed using a magnetic field sensor as shown in Figure 9. In this case, a magnetic field application device is needed to magnetize the label particles. Also, since there must be a linear motion between the LFIA and the magnetic sensor, an actuator such as a linear motor is added. The advantages of magnetic readers are that they are more sensitive and stable than optical signals because they allow the quantification of labeling particles distributed over the entire thickness of the membrane in the assay and are not affected by the background color of the membrane. The target analyte can also be pre-concentrated using magnetism. Readers developed for the quantification of an LFIA using magnetic labeling particles can be classified into magnetoresistive (MR) readers and inductive readers. For the MR reader, a tunnel magnetoresistance (TMR) sensor and a giant magnetoresistance (GMR) sensor [38] are used, and there is also an example of quantitative analysis using an inductive reader [39].

Lei et al. developed a prototype of a reader using a TMR sensor to quantitatively measure the magnetic nanoparticles collected in the LFIA test line. The differential outputs of the two sensors were used for quantification, and the positions and scan speeds of these two sensors were optimized to maximize the measurement performance. The concentration of hCG was quantitatively analyzed, and a magnetic field strength of 0.247 mOe was measured even though the test line was invisible at a low concentration [40]. Park developed a reader using a pair of permanent magnets and a GMR sensor to quantitatively measure LFIA using superparamagnetic nanoparticles as a reporter. As shown in Figure 10, a pair of permanent magnets located on both sides in a diagonal direction were used to simultaneously provide a magnetic field for both the magnetization of the label particles and the biasing of the GMR sensor [41,42]. In order to quantify magnetic particles, there is also a case study in which an output signal was obtained at a common harmonic frequency by exposing it to an alternating magnetic field with two frequencies. This was to improve the signal-to-noise ratio in quantification and to extend the dynamic range [43,44]. Using the developed portable reader and 200nm-diameter magnetic nanoparticles as reporters, PSA was measured up to an LOD of 25 pg/mL [45]. Wang et al. developed a magnetic flux sensor for the quantitative measurement of assays using magnetic nanoparticles as labeling particles and designed the necessary circuit and mechanism. It was applied to hCG and gastrin-17 (G17) measurement to obtain LODs of 0.0098 mIU/mL and 9.1016 pg/mL, respectively [46]. There is also a case study in which multiple sensors were arranged in an array to reduce readout time in quantification of LFIA using GMR sensors. Hall et al. developed a scalable circuit for use with more GMR sensor arrays while maintaining fast read rates. In this study, they were able to scan 64 sensors in less than 4 s using time domain and frequency domain multiplexing and transimpedance amplifiers [38].

Magnetic readers have the following technical difficulties: (1) the relative linear motion between the sensor and LFIA is required for magnetic scanning, (2) the bias of the sensor is required to improve the sensitivity, and (3) the output signal changes very sensitively according to the distance between the sensor and the assay test line. Because of these difficulties in manufacturing magnetic readers, researchers often use commercially available devices as readers for quantitative analysis of the developed LFIA. Cai et al. manufactured poly acrylic acid (PAA)-modified gold magnetic nanoparticles (PGMNs) in order to obtain a stable magnetic signal with a low background noise in quantitative analysis of the PSA. In this study, the portable magnetic reader of MagnaBioSciences, LLC was used as a quantitative reader, and LOD of 0.17 ng/mL was obtained [47].

## 5. Photothermal Readers

Some materials produce diffusion of thermal waves when they absorb light. When these materials are used as LFIA reporters, a reader equipped with a thermal sensor can be used for quantification, as shown in Figure 11. The components of a photothermal reader are almost identical to those used in optical (colorimetric and fluorescence) readers. A light source with a wavelength band suitable for photothermal particle excitation is required. Even label particles present inside LFIA can contribute to quantitative detection, thus overcoming the disadvantages of general colorimetric readers with low sensitivity.

Shirshahi et al. applied the photothermal properties of reduced graphene oxide (rGO) to E-coli O157:H7 quantification. The LOD was 5 × 10^5^ CFU/mL and the sensitivity was improved by about 10 times compared to the conventional colorimetric reader [5]. Ojaghi et al. applied thermophotonic lock-in imaging (TPLI) to improve the signal-to-noise ratio of a photothermal reader using an 808 nm laser and an IR camera. While the detection threshold of the colorimetric method was 2 mIU in hCG quantification, the designed TPLI method was 0.2 mIU [48]. Hu et al. developed a calorimetric immunoassay platform that can quantitatively analyze target substances using the photothermal effect of gold nanocages. A near infrared (NIR) laser source and IR camera were used for the quantification of AFP and zearalenone (ZEN). The developed system showed improved performance compared to the qualitative detection in terms of sensitivity, correlativity, and recovery [49].

## 6. Electrochemical Readers

Electrochemical reader measures changes in voltage, current, or impedance that appear on label particles such as metal nanoparticles or oxidizing enzymes by electron transfer process. Electron transfer processes occur when electrons move from an atom or molecule to another chemical substance. For these electrical measurements, electrodes must be connected to the assay and are usually designed on the assay side using appropriate materials. Figure 12 shows a conceptual diagram of electrochemical readers. An electrical sensor measures electrical properties and an actuator provides the constant voltage or current required for the measurements.

Min et al. developed a disposable kit and electrical detection system that can measure sepsis-related biomarkers (i.e., cytokine interleukin-3 (IL-3)) in blood samples. The electrical detection system measures the current by enzymatic reaction and consists of a potentiostat, microcontroller, signal converter, and amplifier. For the collection and storage of measurement data, a smartphone application was developed. The IL-3 measurement performance was LOD of 10 pg/mL and detection time was 1 h [50]. Zhu et al. developed a quantitative analysis method using a nanozyme that can generate glucose proportional to the amount of DNA oxidative damage biomarker (i.e., 8-hydroxy-2-deoxy guanosine (8-OHdG)) or PSA in conventional LFIA. When sucrose was added to a test line formed through a general antigen–antibody reaction, it was converted into glucose by an enzymatic reaction. This is a unique type of quantification method in which the amount of glucose produced was measured using a commonly used personal glucose meter [51].

## 7. Dual-Signal Readers

For example, the photothermal particles used in the LFIA have a specific color, so a colorimetric method can also be applied, although it is not intended for the original purpose. There are cases where two or more measurement principles were applied to complement each other’s shortcomings. Su et al. analyzed furazolidone qualitatively and quantitatively using MnO_2_-Au as colorimetric and photothermal label particles (i.e., dual-signal biosensor) as shown in Figure 13. In the case of photothermal readout, the test line was irradiated with light with a wavelength of 808 nm and a power of 2 W/cm^2^ for 5 min, and thermal images were acquired using a handheld infrared imager. LODs of 1 ng/mL using the colorimetric method and 0.43 ng/mL using the photothermal method were obtained [52]. This dual-signal biosensing enabled rapid diagnosis via colorimetric readout and, if necessary, quantitative analysis via photothermal measurement.

Fang et al. used magnetic nanobeads (MNBs) to measure sulfamethazine in milk. MNBs performed the following three functions: (1) reduction in the matrix effect, (2) qualitative analysis of sulfamethazine through colorimetric readout of MNBs at concentrations above the cut-off value (25 ng/mL), and (3) quantitative analysis through fluorescence intensity measurement of MNBs at concentrations below the cut-off value. The linear range by fluorescence measurement was 0.033–33 ng/mL, and the LOD was 0.026 ng/mL. On the other hand, the linear range by the colorimetric method was 1–100 ng/mL, and the limit of detection was 0.71 ng/mL [53]. Zhu et al. developed LFIA for measuring 8-OhdG by adding electrochemical reading to the basic functions of a conventional strip (i.e., colorimetric strip) to increase reliability and accuracy. The LOD performance was 2.07 ng/mL through the colorimetric method and 3.11 ng/mL by the electrochemical readout using carbon nanotube paper as the working electrode [54]. In this case, although the LOD performance of the LFIA applying two different measurement principles is similar to each other, it has the advantage of increasing the reliability of measurement results.

In some cases, a smartphone was used as a dual-signal reader. Hou et al. developed a smartphone-based dual-mode imaging system for quantitative detection of lateral flow test strips using color and fluorescent labels. Using the Sobel operator algorithm, it was possible to improve the ability to distinguish the test line from the background of the membrane. In the case of fluorescence measurement, measurement sensitivity was improved by using only a specific component (i.e., red channel) of the assay color image. The developed system was applied to quantitative measurement of hCG and CEA and showed LOD performance of 2.3 mIU/mL and 0.037 ng/mL, respectively [55].

## 8. Readers for Quantitative COVID-19 Analysis

After coronavirus disease 2019 (COVID-19), caused by severe acute respiratory syndrome coronavirus 2 (SARS-CoV-2), was first identified in China in 2019, the World Health Organization (WHO) declared a global pandemic in 2020. The number of total deaths from COVID-19 so far exceeds 6 million [56,57]. Accordingly, the demand for a rapid and accurate primary-screening method for COVID-19 has exploded. Diagnosis of COVID-19 using enzyme-linked immunosorbent assay (ELISA) or reverse transcriptase-polymerase chain reaction (RT-PCR) is expensive, takes a long time to obtain results, and requires well-trained and professional operators. Therefore, there is an urgent need for an inexpensive, short-test-time, and easy-to-manipulate LFIA-based COVID-19 diagnostic method, and its commercialization is actively underway [58]. The sensitivity of LFIA-based diagnosis approached 100% 3 weeks after the onset of symptoms, but only 61–88% after 10 days of symptom onset. Therefore, it is necessary to improve the performance of related diagnostic devices and develop a system capable of quantitative analysis [59,60]. The LFIA for detecting COVID-19 is based on gene detection, antigen detection, or antibody detection. In this section, quantitative analysis systems used for the diagnosis of COVID-19 were reviewed by dividing them into each diagnostic method.

Yu et al. developed a fluorescent labeling particle-based LFIA for detecting RNA-dependent RNA polymerase (RdRp), open reading frame 3 accessory (ORF3a), and nucleocapsid (N) genes related to SARS-CoV-2. A commercially available reader (BMT 1D Scanner, Biometrix Technology Inc., Chuncheon, Korea) was used for the analysis of the fluorescence signal of the assay, and the LOD was 10 copies/test. Measurements of quantification sensitivity were performed at 1000, 100, 10, and 0 copies/test. Detection was possible within 30 min using LFIA, but pretreatment including RT-PCR was required before LFIA application [61]. Although the assay used was not of the LFIA type, Ning et al. implemented a system for point-of-care diagnosis of COVID-19 using saliva samples. Clustered regularly interspaced short palindromic repeats (CRISPR) associated protein 12a (CRISPR-Cas12a) activity was used to amplify the viral amplicon signal, and for this purpose, a smartphone-based fluorescence microscopy device including a laser diode was developed. Using the developed system, a linear range of 1 to 105 copies/uL and an LOD of 0.38 copies/uL were obtained, which exceeded the performance of RT-PCR [62]. There is also a case of development of LFIA that does not require an amplification procedure to detect SARS-CoV-2 ribonucleic acid (RNA). S9.6-monoclonal-antibody-labelled europium-chelate-based fluorescent nanoparticles were used to induce RNA binding. A fluorescent reader was used to detect the fluorescence signal of the test line formed in the assay, and the ratio of the fluorescence intensity between the test line and the control line was used for quantification to improve precision and reproducibility. To secure the sensitivity of the system, only the signal amplification of the fluorescent nanoparticles was sufficient without amplification of the target RNA itself, and the assay time was less than 1 h. All types of samples such as throat swabs and sputum showed 100% sensitivity and 99% specificity compared to RT-PCR results [63].

Compared to the gene and antibody-measurement-based COVID-19 diagnostic method, antigen-based quantitative analysis has relatively few cases of research or commercialization. However, antigen-based COVID-19 diagnostic methods are essential for the early diagnosis of COVID-19. Diao et al. quantitatively analyzed nucleocapsid protein using a fluorescence immunochromatographic assay in nasopharyngeal swab samples and urine samples for SARS-CoV-2 antibody detection. An immunofluorescence analyzer of Bioeasy Biotechnology Co., Ltd. (Shenzhen, China) was used for the analysis of the assay, which was a reader indicating positive or negative results by comparing the sensor output value with a predetermined criterion. In the case of using a nasopharyngeal swab and urine samples, the concordance rates in the nucleic acid test were 100% and 73.6%, respectively. COVID-19 identification through such antibody detection was possible even after 3 days of the first fever [64].

Vaccination is the most effective way to build herd immunity against COVID-19 [65]. However, the effectiveness of these vaccinations is inevitably different for each individual, and the effect may vary depending on the appearance of mutations. Therefore, the measurement of antibodies formed in the body is important to evaluate the effect on the vaccine [66,67,68]. Tong et al. developed a colorimetric polydopamine nanoparticle-based LFIA for the measurement of COVID-19-neutralizing antibody in serum. As a result of quantitative analysis using a smartphone-based reader applied with an artificial intelligence (AI) algorithm, 160 ng/mL LOD and a 625–10,000 ng/mL detection range were obtained, which was superior to the result using gold-nanoparticle-based LFIA. In this study, the AI algorithm was used to accurately identify the test line and the control line and convert the intensity of the detection region to the concentration of the target analyte [69]. As another example, Hung et al. constructed an antibody-measurement-based COVID-19 diagnostic system by combining LFIA using gold colloid as a reporter and a spectrometer for quantitative measurement of test line intensity. Through quantitative analysis using a spectrometer rather than a qualitative analysis to judge with the naked eye, it was possible to secure the sensitivity for diagnosis even when antibody formation was not complete during the initial period of infection. The LOD of the developed system was 186 pg/mL, and it was possible to detect the antibody of the patient 3 days after the onset of symptoms [70].

Systems that have already been commercialized as a reader system for analyzing COVID-19 include Roche Cobas SARS-CoV2, Cepheid Xpert Xpress SARS-CoV2, and Abbott ID Now COVID-19. Their analysis time is 3.5 h, 45 min, and 5 min, respectively. There is a case study comparing the measurement results of the Abbott ID Now COVID-19 system, which has the shortest measurement time, with other systems. Thirty percent of nasopharyngeal swabs in viral transport media samples that were positive in Cepheid’s device showed negative results in Abbott’s system. In addition, 45% of samples using dry nasal swabs, which were also tested as positive using Cepheid’s device, showed negative results in Abbott’s system [71].

Table 2 classifies the cases mentioned in this review by reader type and compares each target analyte and quantitative measurement performance. Additionally, Table 3 shows the list of commercially available quantitative analysis readers mentioned in this review, their manufacturers, and characteristics such as measurement types.

## 9. Conclusions

The LFIA allows the target analyte to bind or react with its receptor. Then, qualitative or quantitative analysis is performed using a reporter bound to the receptor. In the case of LFIA for quantitative analysis, colorimetric, fluorescent, magnetic, photothermal, electrochemical, or dual-signal readers can be used depending on the type of reporter. Considering the simplicity of the reader configuration, the cost of production, the size of the reader, and the performance and time required for quantitative analysis, each has its own advantages and disadvantages. In general, optical readers such as colorimetric readers are advantageous for portability and low cost, but their performance is inferior to other readers. Dual-signal readers are sometimes used to simultaneously take advantage of each other. Due to the recent COVID-19 pandemic, the demand for a system that quantitatively analyzes related biomarkers is increasing. Various research cases have been published and several quantitative analysis system products have been launched. However, this also exposes the limitations of LFIA-based point-of-care diagnosis, and the discrepancy in results between commercially available quantitative analysis systems is one of them. As with all technological developments in the past, we believe that efforts to solve these problems of the quantitative LFIA system will continue through the application of new reader technology or improvement in existing technologies.

## Figures and Tables

**Figure 1 sensors-22-07398-f001:**
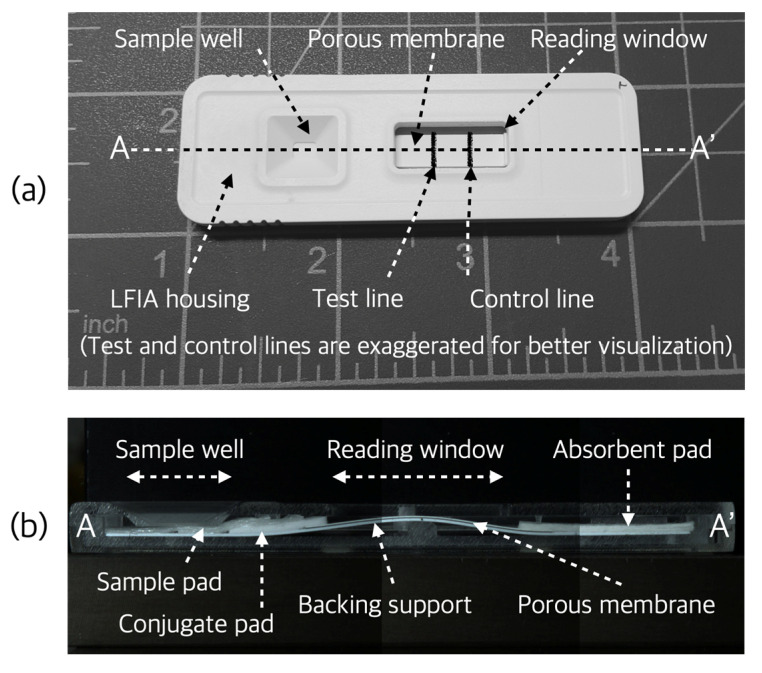
A photograph of the LFIA combined with the housing (**a**) and a photograph of section A-A’ of (**a**,**b**). In (**b**), the LFIA of the reading window area is intentionally bent for the experiment of other research subjects being conducted by the author, and the general LFIA is not.

**Figure 2 sensors-22-07398-f002:**
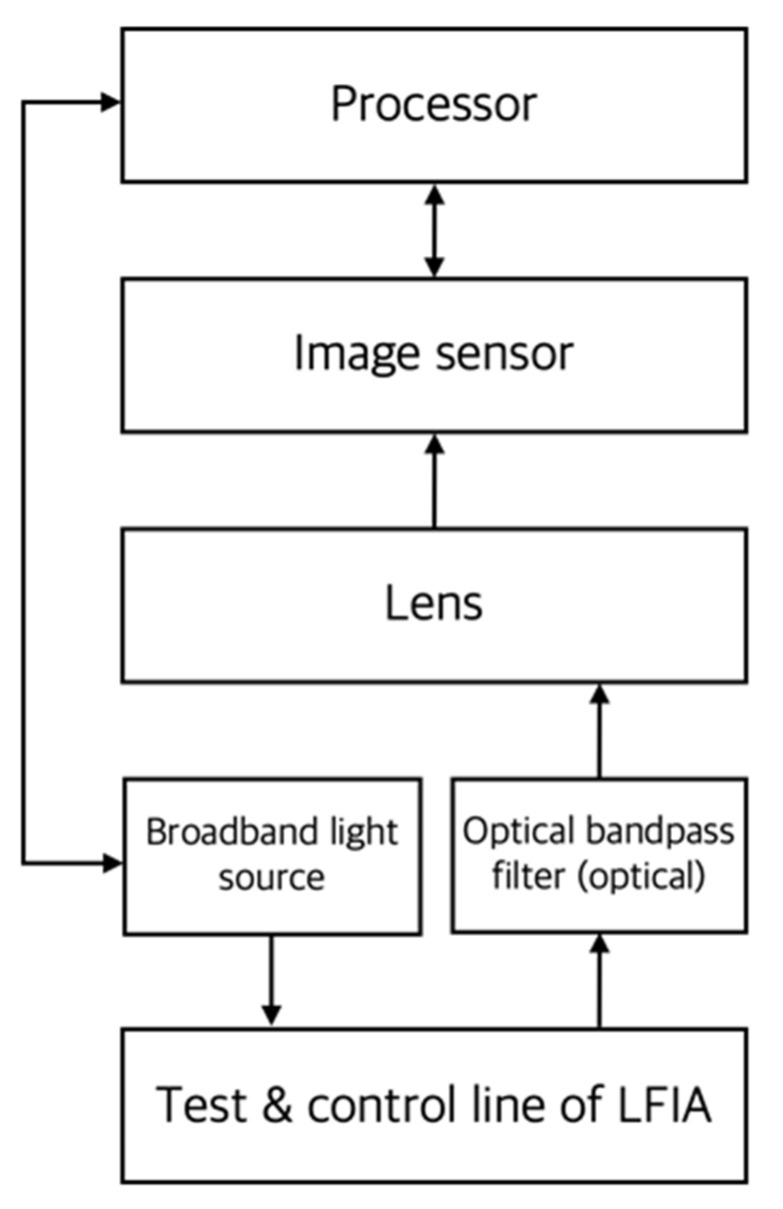
Conceptual diagram of the main components of a colorimetric reader. In the figure, the arrow indicates the direction of the signal or energy transmission. In this case, a broadband light source is usually used for illumination of the LFIA.

**Figure 3 sensors-22-07398-f003:**
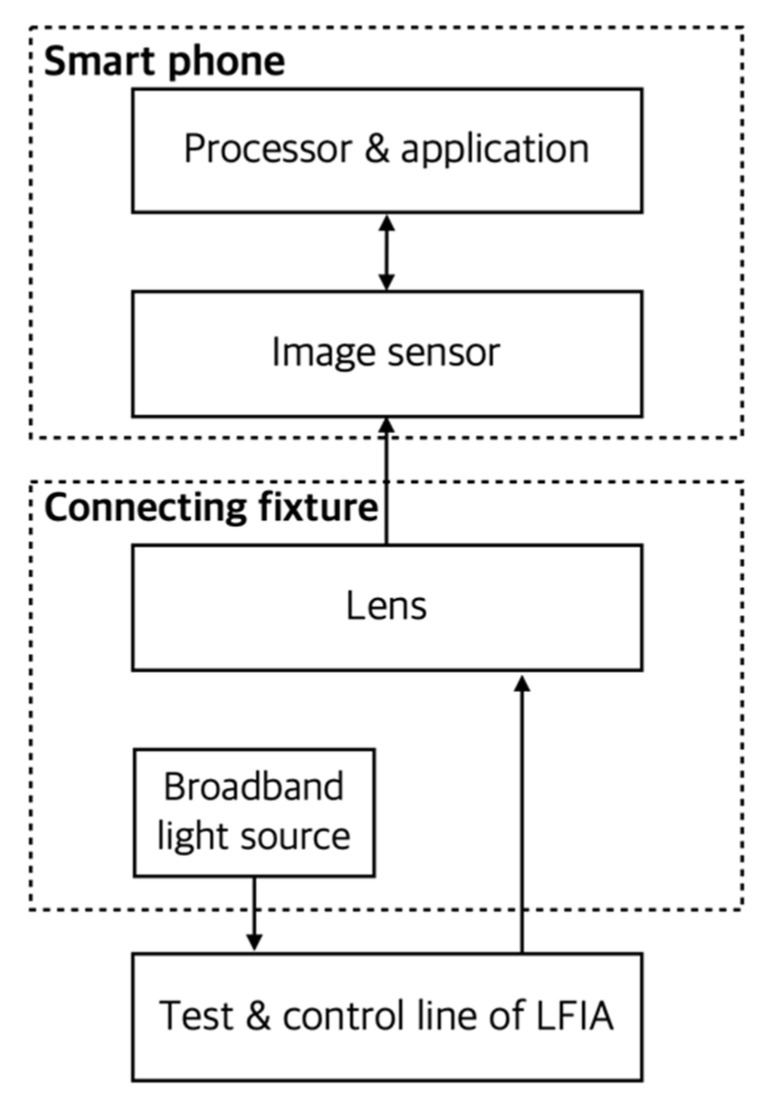
Conceptual diagram of NutriPhone system. In the figure, the arrow indicates the direction of signal or energy transmission. A fixture that connects LFIA and smartphone was made, which included a lens and a light source.

**Figure 4 sensors-22-07398-f004:**
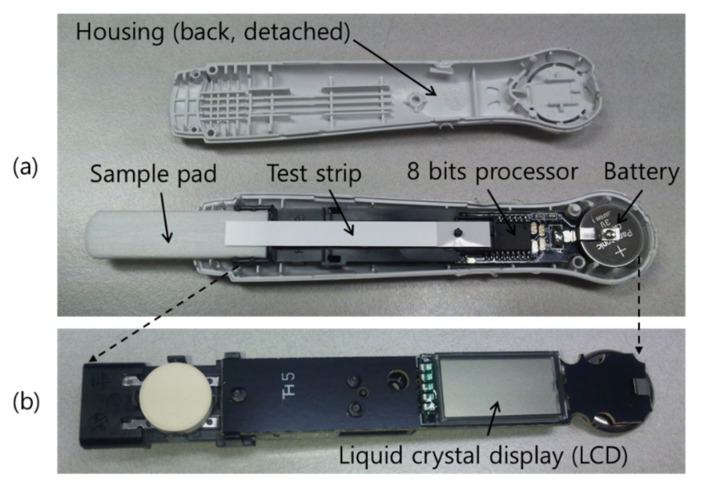
Pictures of disassembled Clearblue^®^. (**b**) is a picture taken by turning over the part indicated by the dotted arrow in (**a**), and the LCD for text display of the test result is shown.

**Figure 5 sensors-22-07398-f005:**
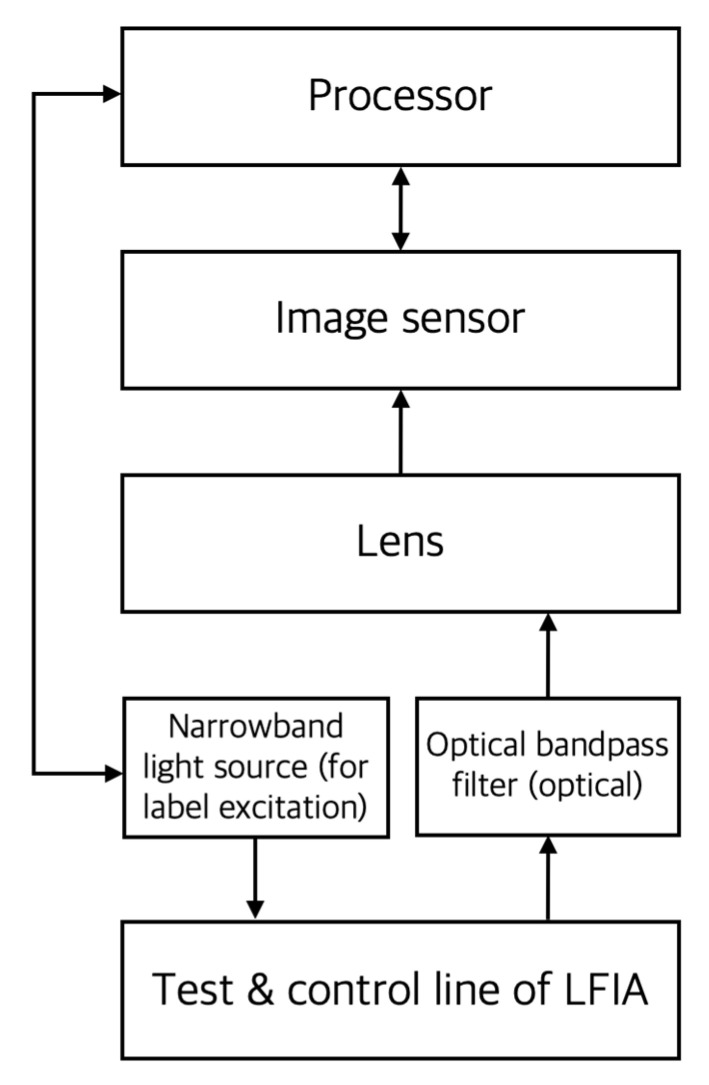
A conceptual diagram of the main components of a fluorescence reader. In the figure, the arrow indicates the direction of signal or energy transmission.

**Figure 6 sensors-22-07398-f006:**
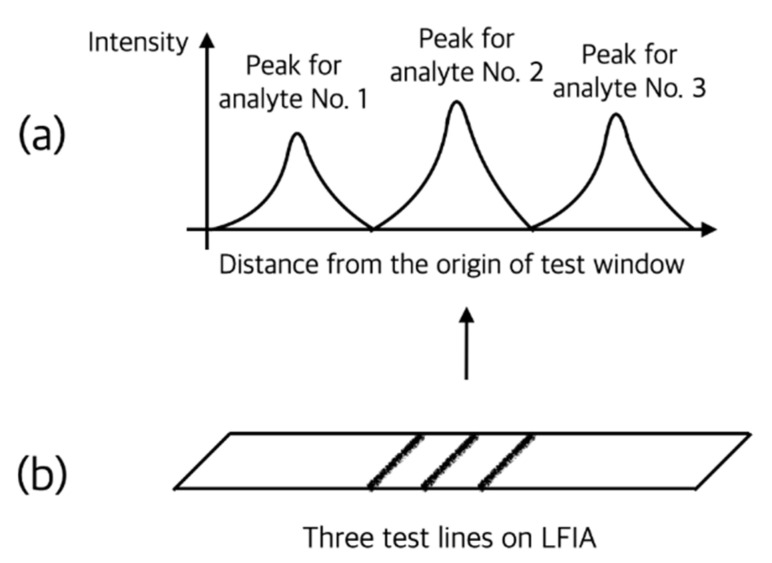
Schematic illustration of LFIA for simultaneous quantification of multiple analytes (**b**) and optical signal output of the reader (**a**). As shown in the figure, signals from three different label particles (i.e., test lines) in one LFIA are spatially separated.

**Figure 7 sensors-22-07398-f007:**
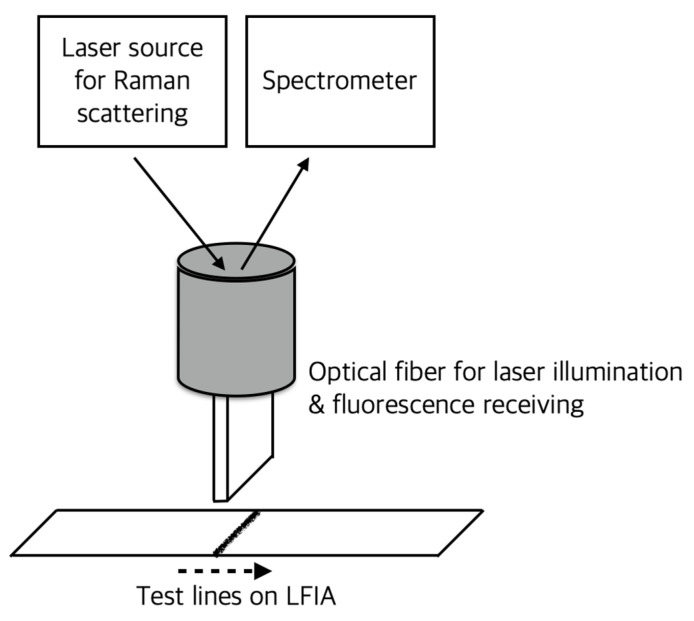
Conceptual illustration of reader for fast scan of SERS-based LFIA. In the figure, the dotted arrow indicates the linear motion of the LFIA. A custom-made optical fiber was fabricated to obtain a scattering pattern of the entire width of the test line (4 mm) in a short time.

**Figure 8 sensors-22-07398-f008:**
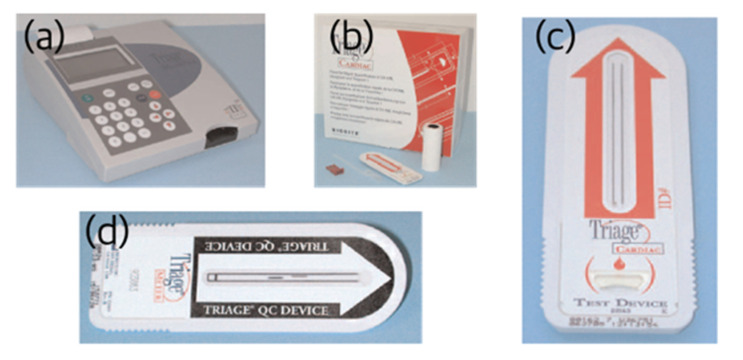
Alere’s Triage^®^ system based on fluorescence immunoassay. Photographs of the reader (**a**), the LFIA used for quantitative analysis (**b**,**c**), and the QC device used for quality control (**d**) are shown.

**Figure 9 sensors-22-07398-f009:**
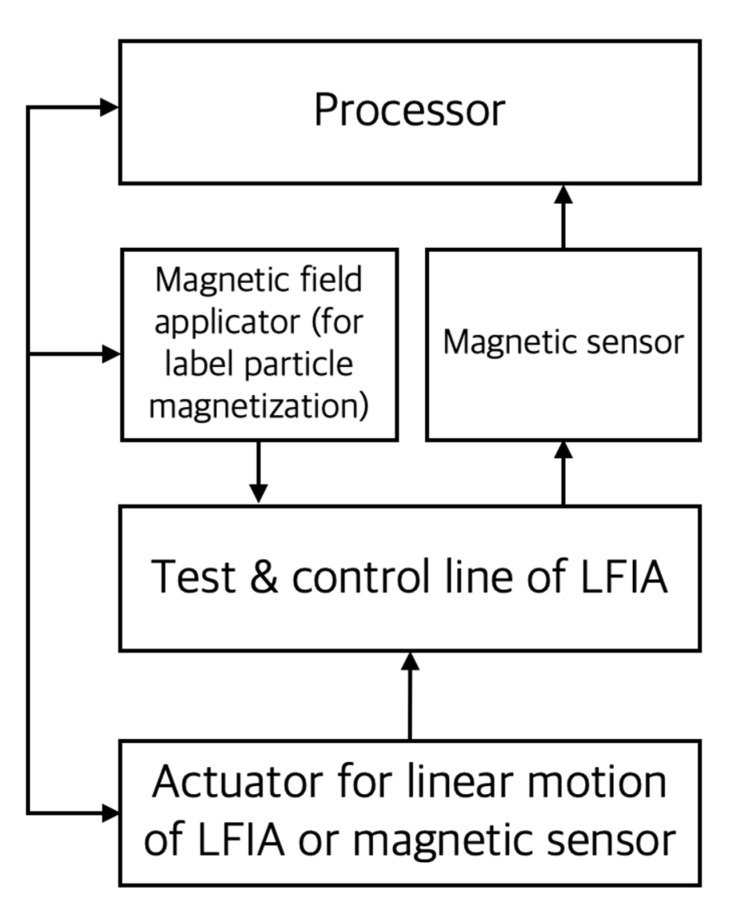
Conceptual diagram of the main components of the magnetic reader. In the figure, the solid arrow indicates the direction of the signal or energy transmission.

**Figure 10 sensors-22-07398-f010:**
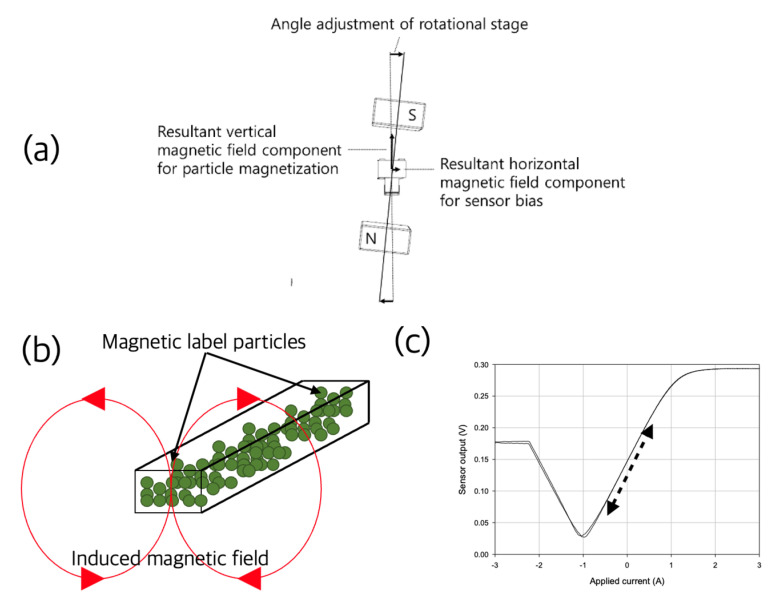
A magnetic-field-applying device utilizing a permanent magnet and its functions. The two permanent magnets are slightly tilted from the vertical line as shown (**a**). The vertical component of the magnetic field formed by this arrangement magnetized the magnetic label particles as shown in (**b**), and the horizontal component biased the sensor, and the output signal was detected in the area indicated by the dotted arrow in (**c**).

**Figure 11 sensors-22-07398-f011:**
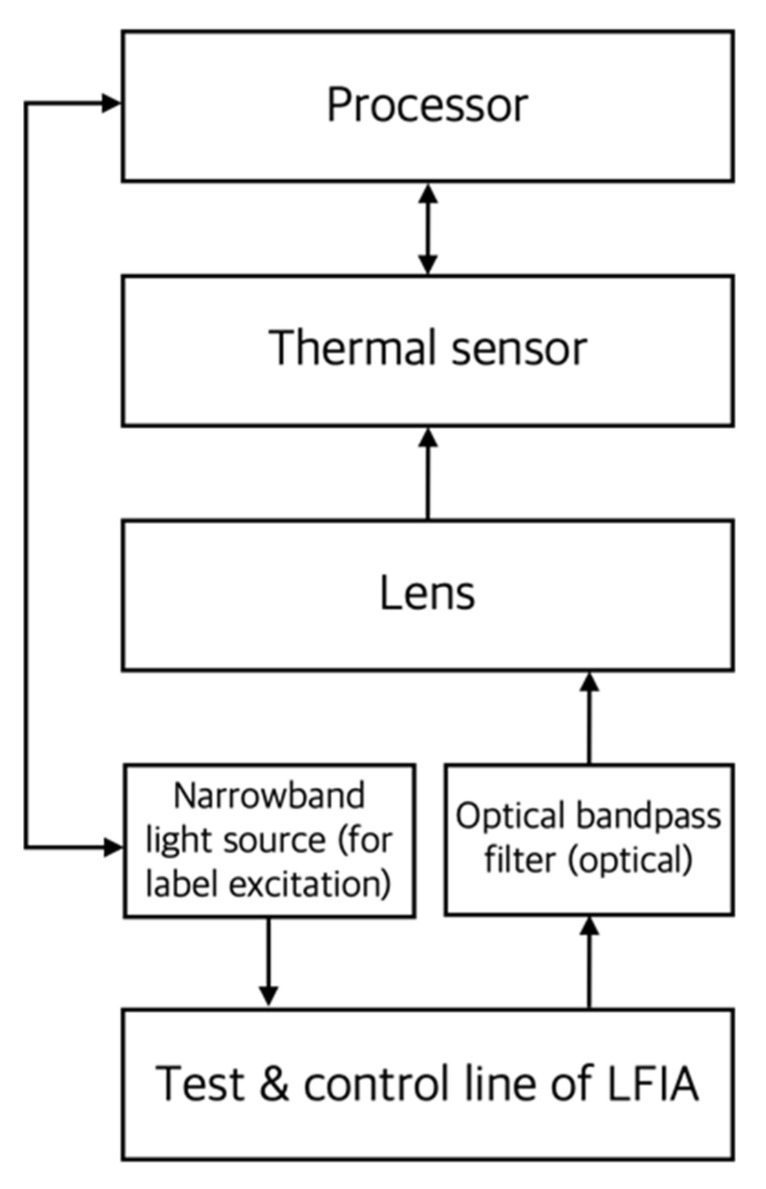
Conceptual diagram of the main components of a photothermal reader. In the figure, the arrow indicates the direction of signal or energy transmission.

**Figure 12 sensors-22-07398-f012:**
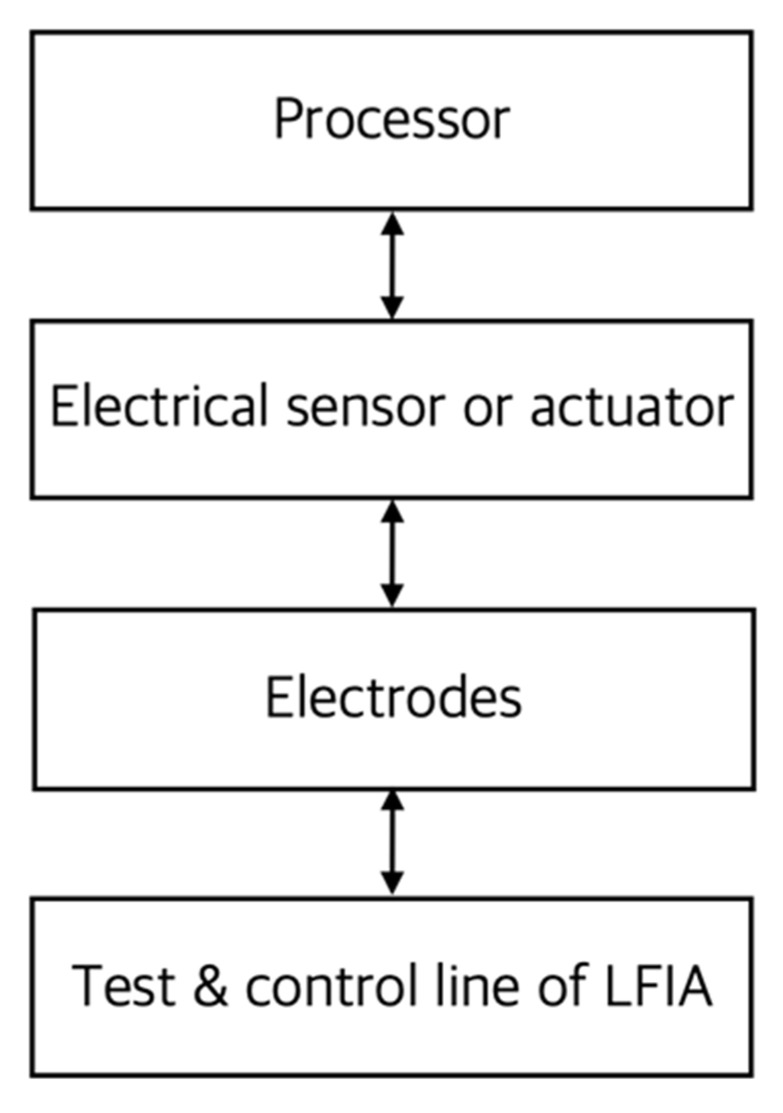
Conceptual diagram of the main components of an electrochemical reader. In the figure, the arrow indicates the direction of signal or energy transmission. The electrode transmits a signal or energy between the LFIA and the reader.

**Figure 13 sensors-22-07398-f013:**
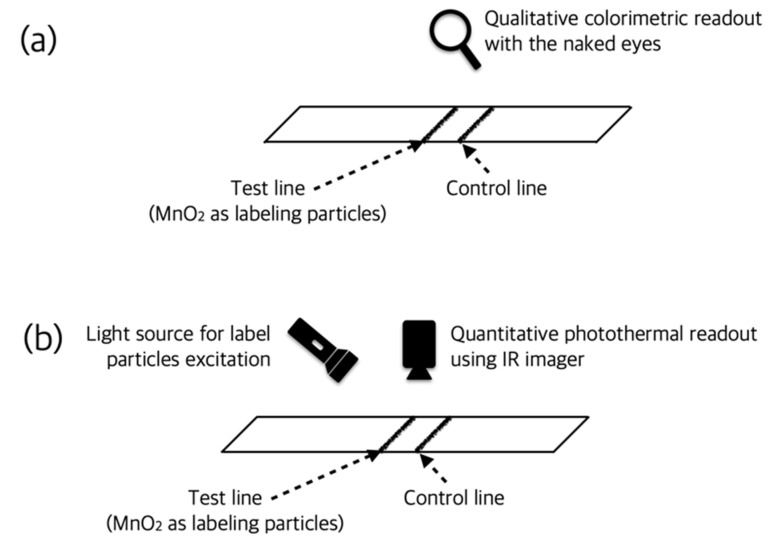
Qualitative colorimetric readout (**a**) and quantitative photothermal readout (**b**) of dual-signal readers. Since the MnO2 label particle used in this study has a specific color and photothermal properties at the same time, it could be used as a reporter in both methods.

**Table 1 sensors-22-07398-t001:** Comparison of reader, label particle, output signal, and sensor used for quantification of a lateral flow immunoassay.

Reader Type	Label Particle	Measured Signal	Sensor
Colorimetric	Colloidal gold or colored monodisperse latex	Contrast or color change	Charge-coupled device or complementary metal–oxide–semiconductor sensor
Fluorescence	Fluorescent monodisperse latex	Fluorescence	Charge-coupled device or complementary metal–oxide–semiconductor sensor
Magnetic	Paramagnetic monodisperse latex or superparamagnetic iron oxide	Magnetic field intensity	Giant magnetoresistive, tunnel magnetoresistive, or inductive magnetic sensor
Photothermal	Graphene oxide or gold nanocage	Thermal waves	Infrared camera
Electrochemical	Metal nanoparticle or oxidizing enzyme	Voltage, current, or impedance	Potentiometer or galvanometer
Dual-signal	Mix of the above	Mix of the above	Mix of the above

**Table 2 sensors-22-07398-t002:** Comparison of measurement target and quantification performance by reader type.

Reader Type	Target Analytes	Range of Concentration Measured and LOD *	Reference
Colorimetric	Creatine kinase-muscle/brain (CK-MB)	0 to 10 ng/mL	[10]
		2 ng/mL	
	*Acidovorax avenae* subsp. *citrulli* (A*ac*)	0 to 1 × 10^7^ CFU/mL	[12]
		1 × 10^5^ CFU/mL	
	Hg(II)	2 to 7 nM	[13]
		2.53 nM	
	Vitamin B_12_	0 to 1107 pmol/L	[14]
		92 pmol/L	
	Spiked serum digoxigenin	0 to 100 nmol/L	[15]
		19.8 nmol/L	
	Cadmium ion	0.16 ng/mL	[16]
	Clenbuterol	0.046 ng/mL	
	Porcine epidemic diarrhea virus	0.055 ug/mL	
	Alkaline phosphatase (ALP)	0.1–150 U/L	[17]
		0.1 U/L	
	Human chorionic gonadotropin (hCG)	6–300 ng/mL	[18]
		3 ng/mL	
	Chromium ions	5–80 ng/mL	[20]
		4.8 ng/mL	
	COVID-19-neutralizing antibody	625 to 10,000 ng/mL	[69]
		160 ng/mL	
	COVID-19 neutralizing antibody	186 pg/mL (with spectrometer)	[70]
Fluorescence	Chloramphenicol	0.3 to 20 ng/mL	[21]
		0.2 ng/mL	
	Iron (ferritin)	10.9 ng/mL	[22]
	Vitamin A (retinol-binding protein)	2.2 ug/mL	
	Inflammation status (C-reactive protein)	0.092 ug/mL	
	Human immunodeficiency virus antibody (HIV Ab)	0.11 NCU/mL	[24]
	Treponema pallidum antibody (TP Ab)	0.62 IU/L	
	Hepatitis C virus antibody (HCV Ab)	0.14 NCU/mL	
	Hepatitis B virus surface antigen (HBsAg)	0.22 IU/mL	
	Alpha-fetoprotein (AFP)	0.01 ng/mL	[9]
	Carcinoembryonic antigen (CEA)		
	Prostate-specific antigen (PSA)		
	Human chorionic gonadotropin (hCG)	1.6 mIU/mL	[26]
	Salinomycin	39 pg/mL	[27]
	Brucellosis	1.6 to 100 IU/mL	[29]
		0.3 IU/mL	
	Mercury ions	5 ppb	[30,31]
	Ochratoxin A	3 ng/mL	
	Salmonella	85 CFU/mL	
	Yersinia pestis	10^4^ to 10^8^ CFU/mL	[37]
		10^4^ CFU/mL	
	RNA-dependent RNA polymerase (RdRp)	10 copies/test	[61]
	Open reading frame 3 accessory (ORF3a), Nucleocapsid (N) genes		
	SARS-CoV-2 RNA	1 to 105 copies/uL	[62]
		0.38 copies/uL	
	Nucleocapsid protein	Qualitative	[63]
Magnetic	Human chorionic gonadotropin (hCG)	25 mIU/mL	[40]
	Lipopolysaccharide of *Francisella tularensis*	0.1 ng/mL	[43,44]
	F1 capsular antigen of *Yersinia pestis*		
	Prostate-specific antigen (PSA)	0.01 to 100 ng/mL	[45]
		25 pg/mL	
	Human chorionic gonadotropin (hCG)	0.0098 mIU/mL	[46]
	gastrin-17 (G17)	9.1016 pg/mL	
	Prostate-specific antigen (PSA)	1 to 128 ng/mL	[47]
		0.17 ng/mL	
Photothermal	*E-coli* O157:H7	5 × 10^5^ to 5 × 10^7^ CFU/mL	[5]
		5 × 10^5^ CFU/mL	
	Human chorionic gonadotropin (hCG)	0 to 16 mIU/mL	[48]
		0.2 mIU/mL	
	Alpha-fetoprotein (AFP)	0.25 ng/mL	[49]
Electrochemical	Cytokine interleukin-3 (IL-3)	10 pg/mL	[50]
	8-hydroxy-2-deoxy guanosine (8-OHdG)	0.1 to 100 ng/mL and 0.23 ng/mL (for 8-OHdG)	[51]
	Prostate-specific antigen (PSA)	1 to 100 ng/mL and 1.26 ng/mL (for PSA)	
Dual signal	Furazolidone	1 ng/mL (for colorimetric)	[52]
		0.43 ng/mL (for photothermal)	
	Sulfamethazine	0.033 to 33 ng/mL and 0.026 ng/mL (for fluorescence)	[53]
		1 to 100 ng/mL and 0.71 ng/mL (for colorimetric)	
	8-hydroxy-2-deoxy guanosine (8-OHdG)	2.07 ng/mL (for colorimetric)	[54]
		3.11 ng/mL (for electrochemical)	
	Human chorionic gonadotropin (hCG)	2.3 mIU/mL	[55]
	Carcinoembryonic antigen (CEA)	0.037 ng/mL	

* Data given as ranges represent linear intervals, otherwise it represents limit of detection.

**Table 3 sensors-22-07398-t003:** List and features of commercial quantitative analysis readers mentioned in this review.

Product Name or Model No.	Manufacturer	Reader Type [Reference]	Remarks
Clearblue^®^	Swiss Precision Diagnostics GmbH	Colorimetric [19]	Optoelectronics (i.e., sensors, LEDs, battery, etc.) that measure the intensity of the test line are integrated into the assay device.
AESE-Quant Portable and Benchtop Lateral Flow Reader	DCN Inc., Irvine, CA, USA	Colorimetric [20]	General-purpose commercial colorimetric reader
AFS-1000	Guangzhou Lanbo Biotechnology Co., Ltd., Guangzhou, China	Fluorescence [27]	General-purpose commercial fluorescence reader
Triage^®^ System	Alere Inc., Waltham, MA, USA	Fluorescence [33]	An immunoassay system for the quantitative analysis of multiple analytes in blood or urine.
Portable magnetic reader	MagnaBioSciences, LLC, San Diego, CA, USA	Magnetic [47]	General-purpose magnetic reader
BMT 1D Scanner	Biometrix Technology Inc., Chuncheon, South Korea	Fluorescence [61]	General-purpose commercial fluorescence reader
An immunofluorescence analyzer	Bioeasy Biotechnology Co., Ltd., Shenzhen, China	Fluorescence [64]	General-purpose commercial fluorescence reader

## Data Availability

Not applicable.

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
