# Peer review of "Lateral Flow Immunoassay Reader Technologies for Quantitative Point-of-Care Testing"

_sensors, 2022, doi:10.3390/s22197398_

Round 1

Reviewer 1 Report

Comments to the Authors

Title “Lateral Flow Immunoassay Reader Technologies for Quantitative Point-of-care Testing”

This manuscript reported lateral flow immunoassay reader technologies for quantitative point-of-care testing. Multiple platforms based on different sensors, for example, optical, magnetic field, photothermal, and electrochemical sensors, are analyzed and compared based on their characteristics, advantages, and disadvantages. Moreover, some special technologies, for example, Raman- or dual signal-based methods are discussed based on their unique characteristics. For each platform, the framework of the detection system is discussed in details. And meanwhile, different labeling methods for different platforms are discussed as well. Since this manuscript proposed a thorough, systematic and well-organized review about lateral flow immunoassay reader technologies, I’d like to recommend its publication.

Following are comments for this review manuscript.

1.     Try to put different technologies into a table for comparison based on different respects.

2.     Try to put different commercialized readers into a table for comparison.

3.     Try to compare the accuracy of quantitative detection based on lateral flow immunoassay reader with that based on other commercialized methods, which will demonstrate the applicability and limitation of lateral flow immunoassay reader in quantitative detection.

Author Response

Reviewer 1

Thank you for your honest and important review comments. The points you mentioned have been revised, and I think the quality of my manuscript has improved a lot as a result, but I am worried that there are some things I have missed because of my lack of it. Below are my actions in response to your comments.

1. Try to put different technologies into a table for comparison based on different respects.

Table 1 showing the characteristics of each measurement principle has been added.

2. Try to put different commercialized readers into a table for comparison.

Table 3 has been added to show a list of the mentioned commercial quantitative analysis readers and their characteristics.

3. Try to compare the accuracy of quantitative detection based on lateral flow immunoassay reader with that based on other commercialized methods, which will demonstrate the applicability and limitation of lateral flow immunoassay reader in quantitative detection.

Table 2 has been added for comparison.

Reviewer 2 Report

I recommend its publication with minor revision and re review as listed below.

1.  Abstract needs more quantitative information. 

2. English needs some improvement.

3. Lateral flow immunoassay (LFIA) 60 years, if possible please include recent developments means improvements in the method 

4. Algorithm was developed for noise reduction of the obtained assay image and image segmentation of ROI regions- more evidence need for this justification 

5.  Must include comparison sensitivity table 

6. Future perspectives missing 

7. such as significantly reducing the detection limit, this sentence confusing 

8. Fluorescence Readers must compare with recent research works 

9. Magnetic particles can be used for modification or for what 

10. Electrochemical Readers in this part must include regarding electron transfer process

11. a fluorescent labeling particle- please mention sensitivity 

12. In introduction, there is a less information about importance of the article 

13. Authors are encouraged to incorporate important in keywords list.

14.  Authors are encouraged to provide noise free – please give more justification

15. Recheck all the abbreviations

16. Figures quality is too low?

17. Material used is missing?

Author Response

Reviewer 2

Thank you for your honest and important review comments. The points you mentioned have been revised, and I think the quality of my manuscript has improved a lot as a result, but I am worried that there are some things I have missed because of my lack of it. Below are my actions in response to your comments.

1. Abstract needs more quantitative information.

The submitted paper is a review, and there is no experimental data conducted by the author. If you tell me the specific type of quantitative information that must be included in the abstract, I’ll be happy to add it.

2. English needs some improvement.

Fixed some typos and flawed English expressions.

3. Lateral flow immunoassay (LFIA) 60 years, if possible please include recent developments means improvements in the method

If you are referring to the up-to-dateness of the resources used in the manuscript, I reviewed the latest trends for each quantitative analysis technology as you can see in the reference list. If I misunderstood your suggestion, please be more specific and I'll be happy to add it.

4. Algorithm was developed for noise reduction of the obtained assay image and image segmentation of ROI regions- more evidence need for this justification

The following sentences have been added, and the added contents are in red in the manuscript.

“The degree of improvement was evaluated by comparing peak signal to noise ratio (PSNR) and contrast of between-class variance (CBCV) values before and after application of the developed algorithm. These values are indicators of image quality and image segmentation performance, respectively.”

5. Must include comparison sensitivity table

Table 2 has been added to compare quantification performance indicators for each technology.

6. Future perspectives missing

The closing sentences of the conclusion (i.e., line 564-569) and some of the introduction (line 57-61) are considered forward-looking statements.

7. such as significantly reducing the detection limit, this sentence confusing

The sentence has been deleted.

8. Fluorescence Readers must compare with recent research works

Table 2, which describes all reader technologies as well as fluorescence technology, has been added to enable comparison.

9. Magnetic particles can be used for modification or for what

I don't quite understand what you're suggesting from the above comments alone. Please be more specific. Can you tell me the relevant part of the text (number of lines) along with what needs to be supplemented?

10. Electrochemical Readers in this part must include regarding electron transfer process

The following sentences have been added, and the added contents are in red in the manuscript.

“Electrochemical reader measures changes in voltage, current, or impedance that appear on label particles such as metal nanoparticles or oxidizing enzymes by electron transfer process. Electron transfer processes occur when electrons move from an atom or molecule to another chemical substance.”

11. a fluorescent labeling particle- please mention sensitivity

The following sentences have been added, and the added contents are in red in the manuscript.

“Measurements of quantification sensitivity were performed at 1000, 100, 10, and 0 copies/test.”

12. In introduction, there is a less information about importance of the article

The following sentences have been added, and the added contents are in red in the manuscript.

“Trend analysis or performance comparison described for each type of reader will be helpful for researchers conducting related research in selecting an appropriate quantitative analysis method and verifying the performance of the developed system.”

13. Authors are encouraged to incorporate important in keywords list.

More important keywords have been added, and the added content is displayed in the text in red.

14. Authors are encouraged to provide noise free – please give more justification

The following sentences have been added, and the added contents are in red in the manuscript.

“The degree of improvement was evaluated by comparing peak signal to noise ratio (PSNR) and contrast of between-class variance (CBCV) values before and after application of the developed algorithm. These values are indicators of image quality and image segmentation performance, respectively.”

15. Recheck all the abbreviations:

I checked the abbreviations again and made sure nothing was missing. Items for which the full term is presented once are described only as an abbreviation thereafter.

16. Figures quality is too low?

Relatively poor quality figures have been increased in resolution.

17. Material used is missing?

The submitted paper is a review, and there are no experiments using the material. If there is any information you think is necessary, please be specific. I will be happy to add relevant information.